# Langerhans Cell Histiocytosis-Associated Pulmonary Adenocarcinoma: A Word of Caution during Molecular Determinations

Laura Melocchi [1],*, Michele Mondoni [2], Umberto Malapelle [3] and Giulio Rossi [1]

1 Pathology Unit, Department of Oncology, Fondazione Poliambulanza Hospital Institute, 25121 Brescia, Italy
2 Respiratory Unit, ASST Santi Paolo e Carlo, Department of Health Sciences, University of Milan, 20142 Milan, Italy
3 Department of Public Health, University of Naples Federico II, 80131 Naples, Italy
* Correspondence: laura.melocchi@poliambulanza.it; Tel.: +39-0303518661

**Abstract:** Background: Smoking habit is a common cause of pulmonary Langerhans cell histiocytosis (PLCH) and lung cancer and both diseases may coexist in the lung and share genetic alterations, such as *V600E BRAF* mutations. We collected a small series of three cases of PLCH-associated lung adenocarcinoma in order to evaluate the molecular setup in both components and underline the critical role of careful tissue selection for predictive molecular driver testing. Methods: Three cases of PLCH-associated adenocarcinoma were collected from consultation files. Clinical data from referring physicians and clinical data were obtained. The surgical biopsies were tested by immunohistochemistry and molecular analysis after separate dissection of adenocarcinoma cells and Langerhans histiocytes. Results: There were three active smoking men with a median age at diagnosis of 60.6 years. PLCH was disclosed at imaging during work-up for suspected lung cancer. Molecular analysis revealed *KRAS* (G12C and G13C) mutations in two cases and *V600E BRAF* mutation in one case of PLCH. Immunostaining with the *V600E BRAF* mutation specific primary antibody VE1 correctly recognized *BRAF*-mutated LCH. One case was wild-type in both diseases. Two similar cases were found in the literature, one of which showed a discrepant *KRAS* (G12D) mutation in adenocarcinoma and a *V600E BRAF* mutation in LCH; Conclusions: This case series of PLCH-associated adenocarcinoma underline the possibility to disclose identical genetic alterations in co-existing benign and malignant pathologies, then potentially creating erroneous interpretation of molecular analysis leading to inadequate therapeutic options in case of incorrect diagnostic recognition and inappropriate selection of both components through microdissection.

**Keywords:** Langerhans cell histiocytosis; adenocarcinoma; BRAF; mutations; lung





## 1. Introduction

Pulmonary Langerhans cell histiocytosis (PLCH) is a diffuse, smoking-related interstitial lung disease generally presenting with nodules and/or cysts mainly involving the upper lobes [1]. Some reports have described the synchronous occurrence of lung adenocarcinoma and LCH, hypothesizing that scarring fibrosis generated by long-standing PLCH may increase the risk of developing lung carcinoma [2–11]. In the study by Sadoun et al. [5] the average time of cancer occurrence from LCH was 10.5 years, while patients with lung cancer and LCH were significantly older than those with only PLCH (64.7 years vs. 40.8 years, *p* < 0.01). Nevertheless, smoking has the key role in promoting both PLCH and lung cancer [2–11]. More recently, PLCH has been regarded as a clonal, neoplastic process of peculiar histiocytes (Langerhans cells) co-expressing S100 protein, CD1a and Langerin [12], as confirmed in the last World Health Organization classification of thoracic tumors [13]. Indeed, activating pathogenic mutations in the mitogen-activated protein kinase (MAPK) pathway have been described in PLCH, involving *proto-oncogene B-Raf (BRAF), Kirsten rat*

*sarcoma virus (KRAS)*, *NRAS proto-oncogene (NRAS)* and *mitogen-activated protein kinase 1 (MAP2K1)* genes [12,14,15]. The discovery of the aforementioned driving mutations paves the way for successful targeted therapies in progressing PLCH [11,14].

Notably, Kobayashi and Tojo [16] demonstrated the feasibility of *BRAF-V600E* determination in cell-free (cf)DNA in a small series of adults with high-risk LCH. In 2017, Zhang et al. [17] reported the possibility to detect *BRAF* and *NRAS* mutations in different clones of circulating Langerhans cells from a patient with PLCH, but not in cell-free DNA nor in circulating cells from patients with lymphangioleiomyomatosis (LAM). Heritier et al. [18] demonstrated the presence of circulating cell-free *BRAF p.V600E* mutation in childhood LCH.

Among PLCH-associated lung adenocarcinoma analyzed at the molecular level, Kalchiem-Dekel et al. [9] and Alden et al. [10] described PLCH discovered on surveillance imaging in patients with pre-existing lung adenocarcinoma. In one case the authors demonstrated *BRAF p.V600E* mutation in adenocarcinoma, but not in PLCH [9], while *BRAF-V600E*-mutant PLCH coexisting with *KRAS*-mutant adenocarcinoma was noted in the other case [10].

Here, we report three further patients with lung adenocarcinoma concomitantly arisen in striking contiguity with PLCH nodules and studied at the molecular level in both components. The key role of a careful selection of the two lesions in tissue biopsy is emphasized in order to prevent erroneous determination of predictive markers leading to unnecessary targeted treatment, particularly in the era of liquid biopsies [19].

## 2. Materials and Methods

Three cases of primary lung adenocarcinoma arisen in the context of Langherans' cell histiocytosis from the consultation files of one of the authors were collected from January 2013 to January 2022.

Clinical and imaging data at presentation and follow-up information were obtained from the medical records and the referring physicians. All tumor samples were fixed in 10% buffered formaldehyde solution for 12–24 h and paraffin-embedded according to standard histopathological methods. All cases (routine hematoxylin–eosin-stained slides and immunohistochemical stains) were reviewed using a multiheaded microscope by two pathologists. For immunohistochemical analysis, an automated immunostainer (ULTRA, Ventana Medical Systems/Roche, Tucson, AZ, USA) was used, with the following prediluted antibodies (source Ventana Medical Systems, AZ, USA): TTF1 (8G7G3/1), p40 (BC28), CD1a (EP3622), S100 (polyclonal), ALK (D5F3), ROS1 (SP384), BRAF (VE1). Appropriate positive internal and/or external controls were included in each batch.

Tumor tissue was microdissected from formalin-fixed paraffin-embedded (FFPE) sections under microscope guidance and molecular analysis was performed by matrix-assisted laser desorption/ionization time-of-flight alterations (LungCarta Panel v1.0, Agena Bioscience, San Diego, CA, USA) in two cases, while the last case was investigated using an amplicon-based DNA/RNA NGS assay (Oncomine Focus Assay (Thermo Fisher Scientific, Waltham, MA, USA).

The study was conducted in accordance with the precepts of the Helsinki Declaration; all data were handled anonymously and in accordance with local institutional ethical board protocols. A comprehensive search of the literature on PubMed/Medline variably crossing the words "Langerhans cell histiocytosis" and "X histiocytosis" and "lung cancer" was performed until 30 June 2022 without language restriction.

## 3. Results

The most relevant clinical-pathological and molecular findings of the cases included in the current series are summarized in Table 1. There were three men, with mean age at diagnosis of 60.6 years (range 35–75 years). All the patients were active smokers and discovered Langerhans cell histiocytosis at CT scan during imaging studies performed for

suspected lung cancer. Two patients underwent curative lobectomy with regional lymph nodes dissection, while a pulmonary wedge resection was performed in one case.

**Table 1.** Summary of concurrent pulmonary lung cancer and LCH with the relevant molecular findings.

| Reference | Gender | Age | Smoke | Tumor Stage | Genetic Alterations in Lung Cancer | Genetic Alterations in PLCH | Therapy | Outcome |
|---|---|---|---|---|---|---|---|---|
| Alden et al. | Female | 45 | Current | IIIB | *KRAS* (p.G12D) | *BRAF* (p.V600E) | Neoadjuvant chemotherapy; surgery | Alive (>6 months) |
| Kalchiem-Dekel et al. | Female | 56 | Current | IV | *BRAF* (p. V600E) | None | Chemotherapy + radiotherapy | Lost to follow-up |
| Our case | Male | 35 | Current | IIB | *KRAS* (p. G12C) | *BRAF* (p.V600E) | Surgery; adjuvant chemotherapy | Died of disease (11 months) |
| Our case | Male | 75 | Current | IB | *KRAS* (p.G13C) | None | Surgery; chemotherapy | Died of disease (35 months) |
| Our case | Male | 72 | Current | IV | None | None | Chemotherapy | Died of disease (8 months) |
| Khaliq et al. | Female | 76 | Current | IIA | N.A. | N.A. | Surgery | N.A. |
| Von der Thusen et al. | Male | 14 | Never | IV | *KRAS* (p. G12C) | N.A. | Chemotherapy | N.A. (colonic carcinoma with MUTYH-associated polyposis) |
| Bhardwaj et al. | Female | 28 | Current | IV | None (tested for *EGFR* and *ALK* only) | N.A. | Chemotherapy (cisplatin + pemetrexed) | N.A. |
| Kaya et al. | Female | 28 | Current | IIA | N.A. | N.A. | Surgery | N.A. |
| Ohtsuki et al. | Female | 78 | Never | IB | N.A. | N.A. | Surgery | N.A. |
| Gancer et al. | Male | 70 | Current | IA | N.A. (squamous cell carcinoma) | *BRAF* (p.V600E) | Surgery | Alive (6 months) |

Abbreviations: PLCH, pulmonary Langerhans cell histiocytosis; N.A., not available.

Radiological features showed a diffuse and bilateral interstitial lung disease with several small nodules and irregular cysts of various size, mainly involving the upper lobes with sparing of the costophrenic angles, strongly suggestive of LCH (Figure 1A).

At histology, the cases showed irregular, spiculated nodules centered by an inflammatory process with fibrotic tissue lined at the periphery by a neoplastic growth of glands without cilia with an acinar pattern (Figure 1B,C). The central areas of this nodules showed several histiocytes with moderate cytoplasm and irregular nuclei with grooves in the context of fibrosis and inflammatory infiltrate with lymphocytes and eosinophils (Figure 1D). The histiocytic component expressed CD1a (Figure 1E) and *BRAF* VE1 (Figure 1F). Immunohistochemically, the adenocarcinomas were TTF-1 positive and p40 negative, while *ALK*, *ROS1* and VE1 were negative. All cases of LCH showed CD1a and S100 positive histiocytes with *BRAF* VE1 expression in one case. Molecular analysis of LCH and adenocarcinoma components evidenced distinctive profile in two cases, with adenocarcinoma harboring *KRAS* mutations (*p.G12C* and *p.G13C*) and LCH showing *p.BRAF V600E* in one case (also positive at immunohistochemistry) and wild-type profile in the other. No mutations in both components were detected in the last case. Gene fusions were lacking in all cases.

A diagnosis of pulmonary Langerhans cell histiocytosis-associated adenocarcinoma was made in all cases. All patients died of metastatic disease (ranging from 8 to 35 months).

A review of the literature revealed two cases [9,10] of lung adenocarcinoma associated with LCH in which molecular analysis of both diseases was performed. There were two smoking women (45- and 56-year-old) with an adenocarcinoma harboring *KRAS* mutation (p.G12D) and concomitant *BRAF V600E* mutated LCH, while the other case had *BRAF p.V600E* mutation in the adenocarcinoma and wild-type LCH.

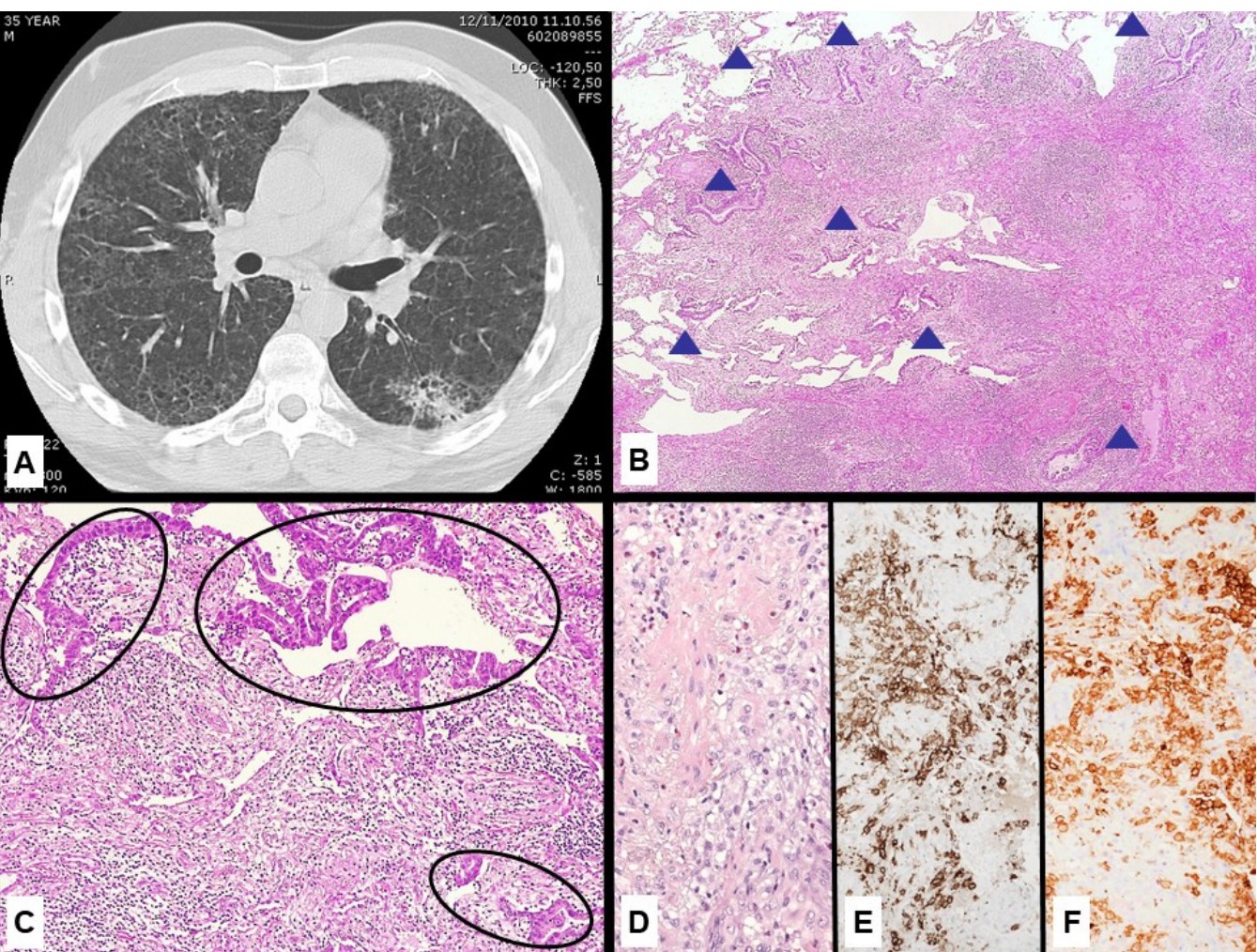

**Figure 1.** Example of PLCH-associated adenocarcinoma (case n. 1 in the current series). (**A**) Chest CT showing an irregular nodule of the left upper lobe in the context of diffuse, bilateral ground glass opacities and cysts; (**B**) Histologic examination of surgical resection revealed an invasive adenocarcinoma surrounding a cellular and fibrotic nodule (hematoxylin–eosin stain); (**C**) At higher magnification, the nodule consisted of an adenocarcinoma with acinar pattern at the periphery (circles) and a central cellular inflammatory infiltrate (hematoxylin–eosin stain) with several Langerhans cells (**D**) expressing CD1a; (**E**), immunohistochemistry and *BRAF V600E*-specific mutation clone VE1; (**F**), immunohistochemistry.

## 4. Discussion

Association between bronchogenic carcinoma and PLCH has been reported in few case series and single reports, but overall, it is rare [1–13]. Sadoun et al. [5] reported 5 bronchogenic carcinoma cases in a series of 93 PLCH patients (5.3%); Vassallo et al. [1] reported 5 primary lung cancers in a series of 102 cases with PLCH (5%), and Lombard et al. [4] reported 4 lung cancer cases of 130 pulmonary PLCH patients (3%). Finally, in a natural history series of 314 PLCH patients from Mayo Clinic [3], only 5 had PLCH and concurrent lung cancer (1.5%), 4 of which were adenocarcinoma and 1 was small cell carcinoma. Among all reported cases of concurrent lung cancer and PLCH, the great majority consists of adenocarcinoma histological type, while just a few are squamous cell carcinomas [13]. Although the concomitant presence of these two entities has been described in the pre-molecular era (Table 1), the most challenging issue in the presence of both lung cancer and PLCH is the possibility of molecular misdiagnosis if pathologists do not pay careful attention in separating tumor tissue from LCH when predictive biomarkers are required.

Overall, only 5 out of 30 cases of lung cancer in PLCH reported in the English literature have been investigated at the molecular level in both components [1–13].

The MAPK signaling pathway is constantly activated in PLCH and *BRAF p.V600E* somatic mutation is present in about 50% of PLCH [14,17–22]. In addition, *MAP2K1* mutations account for 20% of *BRAF* wild-type PLCH lesions [14].

Among several genetic mutations associated with clonal PLCH, the *BRAF p.V600E* activating mutation is the most established [14–22]. Yousem et al. [23] demonstrated *BRAF p.V600E* mutation in all concurrent nodules of 2 out of 5 patients with PLCH, while Kamionek et al. [24] found *BRAF p.V600E* mutations in 8 out of 12 (67%) cellular PLCH and in only 1 out of 14 (7%) fibrotic PLCH. *MAP2K1* or *KRAS* mutations were identified in 4 out of 14 (29%) fibrotic and 3 out of 12 (35%) cellular PLCH. The molecular profile was identical in multiple nodules from the same case.

*BRAF V600E*-specific monoclonal antibody (VE1) is significantly correlated with *p.V600E* mutation detected at molecular level. Roden et al. [25] demonstrated *BRAF V600E* positivity at IHC in 7 out of 25 (28%) PLCH and 6 of them were confirmed by molecular analysis.

Therefore, IHC with clone VE1 may be successfully used in discriminating areas involved by PLCH from adenocarcinoma proliferation, as in one of our cases, helping the microdissection procedures.

Activating mutations in the *BRAF* gene have the potential to contribute to clonal cell expansion [15,19], but *p.V600E* mutation also represent a target for specific inhibitors. Interestingly, the same mutation is also observed in about half of *BRAF*-mutated lung adenocarcinomas and overall, in 1–3% of non-small cell lung cancers, more commonly in current and former smokers [26].

Since a subset of lung adenocarcinoma is associated with PLCH and both diseases may share identical genetic alterations acting as molecular drivers and druggable targets for specific inhibitors, this leads to important considerations on careful tissue selection to submit to molecular analyses.

This issue is particularly challenging since *BRAF p.V600E* mutation may be detected in liquid biopsies from patients with both PLCH and lung adenocarcinoma, preventing distinguishing the exact disease harboring *BRAF p.V600E* mutations [20–22,27,28].

One of our cases had *BRAF p.V600E* mutation in LCH nodules and *KRAS (p.G12C)* mutation in the associated adenocarcinoma, similar to what was observed by Alden et al. [10], while the other two collected cases and one reported by Kalchiem-Dekel et al. [9] did not harbor genetic alterations in LCH but had *KRAS (p.G13C)* and *BRAF p.V600E* mutations only in the adenocarcinoma (Table 1).

Overall, the cases herein demonstrate a little advantage of tissue biopsies over liquid biopsies when lung adenocarcinomas are strikingly admixed with other neoplastic diseases sharing identical genetic alterations, also explaining the unclear response to specific inhibitors.

## 5. Conclusions

This case series of PLCH-associated adenocarcinoma illustrates the complexity of our expanding knowledge regarding the genetic alterations shared by co-existing benign and malignant pathologies [18–22], such as endometriosis [27,28], and well-known clonal hematopoiesis potentially harboring *KRAS* mutations [27–31] that can be misinterpreted as cancer drivers.

Although the concomitant existence of lung adenocarcinoma arising in the context of PLCH is uncommon, these cases are underscored and show overlapping genetic alterations on predictive markers, namely *BRAF V600* mutation, suggesting another possible advantage of tissue biopsies with appropriate microdissection of tumor cells for molecular investigations and subsequent adequate therapeutic management.

Needless to say, a careful examination of clinical-radiologic findings and tissue biopsies are essential to suspect the co-existence of LCH and lung cancer, since both pathologies are significantly related to smoking habit.

**Author Contributions:** Conceptualization, G.R. and L.M.; methodology and formal analysis of data, U.M. and G.R.; investigation and data curation, L.M. and M.M.; writing—original draft preparation, L.M. and G.R.; writing—review and editing, L.M. and M.M.; supervision, G.R. and U.M. All authors have read and agreed to the published version of the manuscript.

**Funding:** This small series of cases did not receive any specific grant from funding agencies in the public, commercial or even not-for-profit sectors.

**Institutional Review Board Statement:** The study was conducted in accordance with the Declaration of Helsinki, all data were handled anonymously and in accordance with local Institutional ethical Review Board protocols.

**Informed Consent Statement:** Informed consent was obtained from all subjects involved in the study while they were alive.

**Data Availability Statement:** Not applicable.

**Conflicts of Interest:** The authors have no disclosure or conflict of interest to declare concerning this work.

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
