# Peer review of "Langerhans Cell Histiocytosis-Associated Pulmonary Adenocarcinoma: A Word of Caution during Molecular Determinations"

_jmp, doi:10.3390/jmp3040024_

Round 1

Reviewer 1 Report

Dears Author's,

It is with great pleasure to recommend your article for publication. It is an interesting volume compared to other publications. 

Congratulation on your work.

Author Response

The authors thank the reviewer so much for the constructive comments, we appreciate it.

Reviewer 2 Report

Melocchi et al., investigated molecular testing of Langerhans’ cell histiocytosis accompanied by lung cancer in three patient samples. Two samples harbored KRAS mutation in which one sample harbored BRAF mutation. The study is interesting and discusses a rare tumor entity. However, sample size is limited to three patients and molecular discovery was limited to two mutations. Authors should create a comprehensive literature review table in order to compare their new results to a larger cohort, which will expose the significance of their findings. 

Author Response

We thank the reviewer for the suggestion. Mutated cases of lung cancer and PLCH reported in literature so far are those presented in Table 1, although the concomitant presence of these two entities have been described in the pre-molecular era. In our view, the most challenging issue in the presence of lung cancer with PLCH is the possibility of molecular misdiagnosis if pathologists do not pose careful attention in separating tumor tissue from LCH when predictive biomarkers are required. In any case, Table 1 has been updated adding also cases of lung cancer and concomitant PLCH lacking molecular determinations in order to better highlight this uncommon but possible condition (See attachment)

Round 2

Reviewer 2 Report

Authors have provided a detailed literature review, which exposes significance of their findings